# Cell autonomous regulation of hippocampal circuitry via Aph1b-γ-secretase/neuregulin 1 signalling

**Pietro Fazzari[1,2,4,5], An Snellinx[1,2,4,5], Victor Sabanov[3], Tariq Ahmed[3], Lutgarde Serneels[1,2,4,5], Annette Gartner[1,2,4,5], S Ali M Shariati[1,2,4,5], Detlef Balschun[3], Bart De Strooper[1,2,4,5,6]***

[1]VIB Center for the Biology of Disease, KU Leuven, Leuven, Belgium; [2]Centre for Human Genetics, KU Leuven, Leuven, Belgium; [3]Laboratory of Biological Psychology, KU Leuven, Leuven, Belgium; [4]Leuven Institute for Neuordegenerative Disorders (LIND), KU Leuven, Leuven, Belgium; [5]University Hospitals Leuven, Leuven, Belgium; [6]Institute of Neurology, University College London, London, United Kingdom

**Abstract** Neuregulin 1 (NRG1) and the γ-secretase subunit APH1B have been previously implicated as genetic risk factors for schizophrenia and schizophrenia relevant deficits have been observed in rodent models with loss of function mutations in either gene. Here we show that the Aph1b-γ-secretase is selectively involved in Nrg1 intracellular signalling. We found that Aph1b-deficient mice display a decrease in excitatory synaptic markers. Electrophysiological recordings show that Aph1b is required for excitatory synaptic transmission and plasticity. Furthermore, gain and loss of function and genetic rescue experiments indicate that Nrg1 intracellular signalling promotes dendritic spine formation downstream of Aph1b-γ-secretase in vitro and in vivo. In conclusion, our study sheds light on the physiological role of Aph1b-γ-secretase in brain and provides a new mechanistic perspective on the relevance of NRG1 processing in schizophrenia.

**\*For correspondence:** bart. destrooper@cme.vib-kuleuven.be

**Reviewing editor**: Eunjoon Kim, Korea Advanced Institute of Science and Technology, South Korea

## Introduction

Schizophrenia (SZ) is a severe disorder that affects neuronal circuits involved in social behaviour and cognitive processes (*Harrison and Weinberger, 2005*; *Lewis and Sweet, 2009*). Increasing evidence suggests that different risk genes interact synergistically to contribute to SZ, mainly by affecting excitatory and/or inhibitory circuitries in the cortex (*Harrison and Weinberger, 2005*; *Lisman et al., 2008*; *Lewis and Sweet, 2009*; *Glausier and Lewis, 2013*). In particular, polymorphisms in Neuregulin 1 (NRG1) have been consistently linked to schizophrenia in different populations (*Stefansson et al., 2002*; *Mei and Xiong, 2008*). The *NRG1* gene encodes more than 30 isoforms that differ in structure, expression pattern, processing and signalling modes which complicates the study of the NRG1 family (*Mei and Xiong, 2008*). Most Ig-Nrg1 isoforms apparently function as diffusible paracrine signals. Conversely, the cysteine-rich domain-(CRD-) containing Nrg1 isoform (also known as Type III Nrg1) is membrane bound and, in addition to canonical forward signalling via ErbB4, can also signal backward via its intracellular domain (Nrg1-ICD) (*Bao et al., 2003*; *Mei and Xiong, 2008*; *Chen et al., 2010a*; *Pedrique and Fazzari, 2010*). Converging studies demonstrate that Nrg1/ErbB4 forward signalling controls the establishment of cortical inhibitory circuits and is implicated in the control of neuronal synchronisation (*Chen et al., 2010b*; *Fazzari et al., 2010*; *Wen et al., 2010*; *Rico and Marin, 2011*; *Cahill et al., 2012*). However, the physiological role of CRD-Nrg1 intracellular signalling, and thus the function of the membrane bound and intracellular domain of Nrg1 remains unclear.

**eLife digest** Schizophrenia affects around 1% of the world's population, with symptoms including hallucinations and delusions, apathy and cognitive impairments. Multiple genes and environmental factors interact to increase the risk of schizophrenia, making the causes of the disease—which can differ between individuals—difficult to disentangle. However, Schizophrenia is known to be associated with a reduction in the number of dendritic spines, the small protrusions that allow brain cells to receive inputs from other brain cells.

One gene that has repeatedly been implicated in schizophrenia is *neuregulin 1* (*NRG1*), which encodes a signalling protein with more than thirty different variants. One of these variants, type III NRG1, is located on the cell membrane. An enzyme called γ-secretase can cleave the 'tail' of this protein, which means that the tail becomes free to move to the nucleus of the cell, where it can alter the expression of genes.

Fazzari et al. have now studied how different γ-secretases interact with type III NRG1 by using genetic techniques to remove a specific part of the enzymes in the brains of mice. The brain cells of these mutant mice contained fewer dendritic spines than mice with normal γ-secretases. However, the number of dendritic spines in the mutant mice could be restored by introducing γ-secretase.

These results are consistent with a model in which mutations that remove the ability of γ-secretases to cleave NRG1 lead to some of the structural and functional changes in the brain that are associated with schizophrenia. An improved understanding of the properties of the various γ-secretases could also lead to the design of safer versions of drugs called γ-secretase modulators that are used to treat Alzheimer's disease.

In analogy to Notch signalling (*De Strooper et al., 1999*), the intracellular part of Nrg1, Nrg1-ICD, is released by intramembrane processing. It is known that γ-secretase activity is responsible for this cleavage (*Bao et al., 2003*; *Dejaegere et al., 2008*; *Chen et al., 2010a*; *Pedrique and Fazzari, 2010*; *Marballi et al., 2012*), but it remains unclear which specific γ-secretase is involved. γ-secretases are a family of intramembrane proteases composed of four different subunits: presenilin (PSEN), anterior pharynx homologue 1 (APH1), nicastrin (NCT), and presenilin enhancer 2 (PEN2) (*De Strooper, 2003*). In the human genome two presenilin (*PSEN1* and *PSEN2*) and two *APH1* (*APH1A* and *APH1B*) are present; thus, at least four different γ-secretase complexes can be generated. One of the major challenges in the γ-secretase field is to understand whether these different γ-secretase complexes have different biological or pathological functions. This question is particularly relevant for understanding the mechanisms that contribute to the molecular pathogenesis of SZ since indirect evidence indicates that NRG1 intracellular signalling might be involved in the risk for this disease. In this regard, a Val-to-Leu mutation in the NRG1 transmembrane domain increases the risk for development of SZ (*Walss-Bass et al., 2006*), impairs intramembrane γ-secretase cleavage of Nrg1 (*Dejaegere et al., 2008*) and abnormal NRG1 processing was found in schizophrenic patients (*Chong et al., 2008*; *Mei and Xiong, 2008*; *Marballi et al., 2012*). Moreover, putative loss of function variants of APH1B, a crucial component of the γ-secretase complex, were found to aggregate with *NRG1* risk alleles in schizophrenia patients (*Hatzimanolis et al., 2013*) and Aph1b-loss of function mutations in rodents display behavioural phenotypes that are relevant for schizophrenia (*Coolen et al., 2005, 2006*; *Dejaegere et al., 2008*).

Rodents have duplicated the *Aph1b* gene during evolution into highly homologous *Aph1b* and *Aph1C*. In order to model human APH1B loss of function, we previously generated double mutant mice for Aph1b and Aph1C (*Serneels et al., 2005*), hereafter referred to as *Aph1bc*$^{fl/fl}$ or *Aph1bc*$^{-/-}$ upon Cre-dependent deletion. We also conditionally targeted the *Aph1a* locus, referred to as *Aph1a*$^{fl/fl}$. We have found that Aph1a-γ-secretase complexes are necessary to activate Notch signalling and genetic deletion of Aph1a leads to a Notch related embryonic lethality (*Serneels et al., 2005*). Conversely, deletion of the Aph1bc-γ-secretase complex does not affect Notch signalling but hampers Nrg1 processing and alters sensory motor gating, working memory and sensitivity to psychotropic drugs, thereby mimicking Nrg1 deficiency and various phenotypes related to schizophrenia (*Coolen et al., 2005, 2006*; *Dejaegere et al., 2008*). However, the selective role of Aph1bc-γ-secretase complexes and the Aph1bc-γ-secretase-dependent processing of Nrg1 in brain wiring and function remained unstudied.

In the current study, we address the question of whether the Aph1bc subunit provides specificity to Nrg1 processing and whether this subunit would indeed be involved in the postulated intracellular signalling of Nrg1. We show here that the Aph1bc-γ-secretase complex controls excitatory circuitry via Nrg1 intracellular signalling. Mice mutant for *Aph1bc* display altered expression of excitatory synaptic markers, impaired synaptic transmission and decreased long term potentiation. Furthermore, single cell deletion of *Aph1bc* in vivo impaired dendritic spine formation which could be rescued by the expression of the Nrg1-ICD. Taken together, these data indicate that Nrg1 intracellular signalling downstream of Aph1bc-γ-secretase complexes promotes in a cell autonomous fashion the formation of excitatory connections in cortical neurons. Hence, our study provides a cellular and molecular mechanistic explanation for the cognitive deficits observed in Aph1bc-γ-secretase deficient mice (*Dejaegere et al., 2008*). More importantly, it provides unique insight into the importance of Nrg1 intracellular signalling in the establishment of functional synapses and the potential aetiological role of misprocessing of NRG1 in the pathogenesis of schizophrenia.

## Results

### Aph1bc loss of function alters the expression of synaptic markers

We reasoned that the behavioural deficits observed in $Aph1bc^{-/-}$ mice (*Dejaegere et al., 2008*) could be due to abnormal development of the brain. To perform the morphometric analysis of control and $Aph1bc^{-/-}$ cortices, we immunolabelled control and mutant brains for Cux1, a marker for layers II/III and IV, and for the panneuronal marker NeuN (*Figure 1A*). We found that Aph1bc deletion did not alter the size of cortical layers or the relative distribution of neurons in different layers (*Figure 1B–C*). Hence, the observed behavioural abnormalities could not be attributed to a gross morphological alteration of the brain structure.

Schizophrenia is characterized by dysfunction in the prefrontal cortex (*Arnsten et al., 2012*), where Aph1bc is highly expressed (*Dejaegere et al., 2008*). In particular, excitatory circuitry is impaired in prefrontal cortex of schizophrenic patients (*Lewis and Sweet, 2009*). Thus, we scrutinized the expression of different pre- and post-synaptic markers in control and $Aph1bc^{-/-}$ prefrontal cortices to test if neuronal connectivity was properly established. Western blot data indicate a small but significant decrease in the expression of the presynaptic markers VGluT1 and Synaptophysin and of the postsynaptic protein PSD95 (*Figure 1D,E*). In addition, we carried out confocal quantitative analysis of the expression of VGluT1 and the postsynaptic marker Homer1. We found that the staining intensity of VGluT1 and Homer1 positive puncta shifted toward lower staining intensities (*Figure 1F–H*). On the other hand, Aph1bc loss of function did not affect the intensity of positive puncta for VGAT, that labels all inhibitory terminals, and for PV, a specific marker for fast spiking interneurons (*Figure 1—figure supplement 1A–C*). In sum, these data suggested that the glutamatergic circuitry is impaired in $Aph1bc^{-/-}$ mice.

### Impaired synaptic transmission and plasticity in Aph1bc deficient mice

The observed synaptic phenotype prompted us to further investigate synaptic function in $Aph1bc^{-/-}$ mice. We showed that Aph1bc is expressed in CA1 and CA3 layers of the hippocampus (*Serneels et al., 2009*). As it was previously reported that deletion of Presenilins in hippocampal pyramidal neurons impairs synaptic plasticity (*Saura et al., 2004*; *Zhang et al., 2009*), we decided to analyse the Schaffer collateral pathway of the hippocampus, in $Aph1bc^{-/-}$ mice. The Input-Output (I/O) curves, that show the field excitatory postsynaptic potentials (fEPSP) in response to stimuli of increasing strength, indicate that baseline synaptic transmission is impaired in $Aph1bc^{+/-}$ heterozygous and $Aph1bc^{-/-}$ homozygous mutant mice as compared to controls (*Figure 2A*). Conversely, paired-pulse facilitation (PPF), a presynaptic form of short term plasticity which reflects release probability, was undistinguishable in control and mutant mice (*Figure 2B*). Next, we studied the relevance of Aph1bc in long-term synaptic plasticity. Induction of long term potentiation (LTP) by three trains of theta burst stimulation is impaired in both heterozygous and homozygous Aph1bc deficient mice compared to controls (*Figure 2C*). Even though reduced basal transmission might in principle interfere with LTP analysis, these results suggest that synaptic plasticity is affected by Aph1bc deletion. Furthermore, we recorded miniature excitatory currents (mEPSCs) which showed normal amplitude but slightly increased inter-event intervals (IEIs) in $Aph1bc^{-/-}$ as compared to control mice (*Figure 2D–F*). Since PPF analysis indicates that Aph1bc deletion does not alter release probability, the increased IEIs in mEPSCs suggest a decrease in the number of release sites that is consistent with the decreased expression of excitatory synaptic markers assessed by immunofluorescence. Altogether, Aph1bc is required for synaptic transmission and long-term synaptic plasticity.

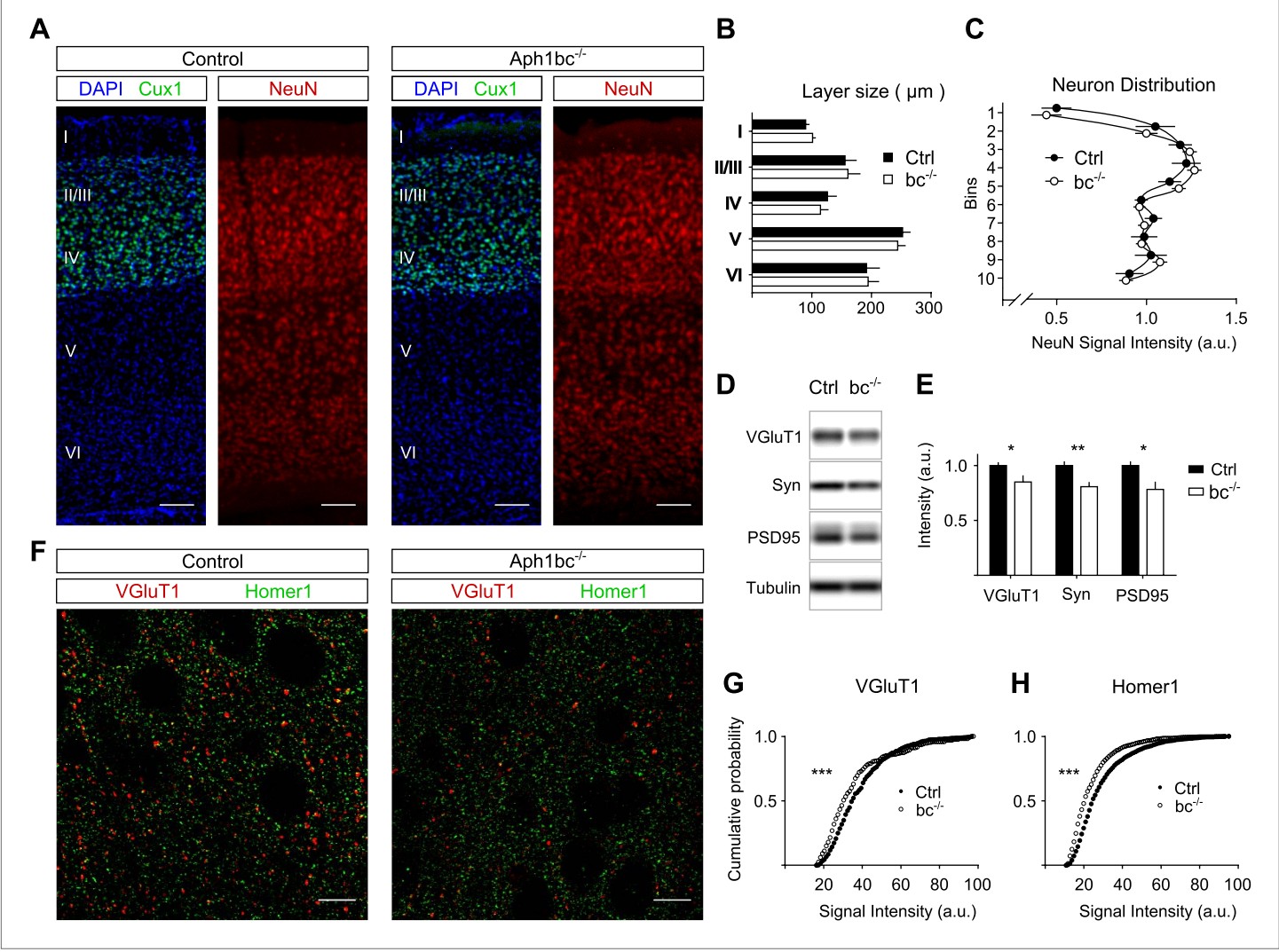

**Figure 1**. Normal cortical layer formation and altered expression of synaptic markers in Aph1bc⁻/⁻ deficient mice. (**A**) Representative pictures of neuronal cortices from Control and *Aph1bc⁻/⁻* null mice at P30 immunostained for the upper layers marker Cux1 and for the pan neuronal marker NeuN. Nuclei were stained with DAPI. Scale bars, 100 µm. (**B**) Quantification of cortical layers size at Bregma −1.4 mm. Ctrl: n = 4; KO: n = 3; Histogram show average ± SD, two way ANOVA. (**C**) Neuronal distribution, as measured by relative NeuN fluorescence intensity along bins ordered from top to bottom, was unchanged in *Aph1bc⁻/⁻* mutant brains at P30. n = 6 sections from three mice; Graph show means ± SD, two way ANOVA. (**D** and **E**) Western blot analysis of synaptic markers VGluT1, Synaptophysin and PSD95 in prefrontal cortex homogenates show decreased expression of these proteins in *Aph1bc⁻/⁻*. n = 9 replicates out of n = 3 mice per group; the histogram shows signal intensity normalized for tubulin signal, means ± SEM, *p<0.05, **p<0.01. VGluT1: Ctrl = 100 ± 3%, *Aph1bc⁻/⁻* = 85 ± 6%; Syn: Ctrl = 100 ± 4%, *Aph1bc⁻/⁻* = 80 ± 4%; PSD95: Ctrl = 100 ± 4%, *Aph1bc⁻/⁻* = 78 ± 7%. (**F**) Representative confocal pictures of layer II/III of prefrontal cortices from control and *Aph1bc⁻/⁻* mice at P30 immunolabelled for the excitatory presynaptic marker VGlut1 and for the excitatory postsynaptic marker Homer1. Scale bar 10 µm. (**G** and **H**) Cumulative probability of VGluT1 and Homer1 puncta intensities in control and *Aph1bc⁻/⁻* mice. VGlut1, n >338 puncta; Homer1, n >1160 puncta; three animals per genotype each, Komolgorov–Smirnov test, ***p<0.001.

The following figure supplements are available for figure 1:

**Figure supplement 1**. *Aph1bc* deletion does not affect the expression of inhibitory synaptic markers.

## Aph1bc-γ-secretase deletion impairs spine formation and is rescued by Nrg1 intracellular signalling in vitro

A reduction in the density of dendritic spines, that receive excitatory inputs, is an hallmark of schizophrenia (*Lewis and Sweet, 2009*). Hence, we investigated the effects of Aph1bc deficiency on the establishment of dendritic spines. We generated hippocampal cultures from *Aph1bc^{fl/fl}*

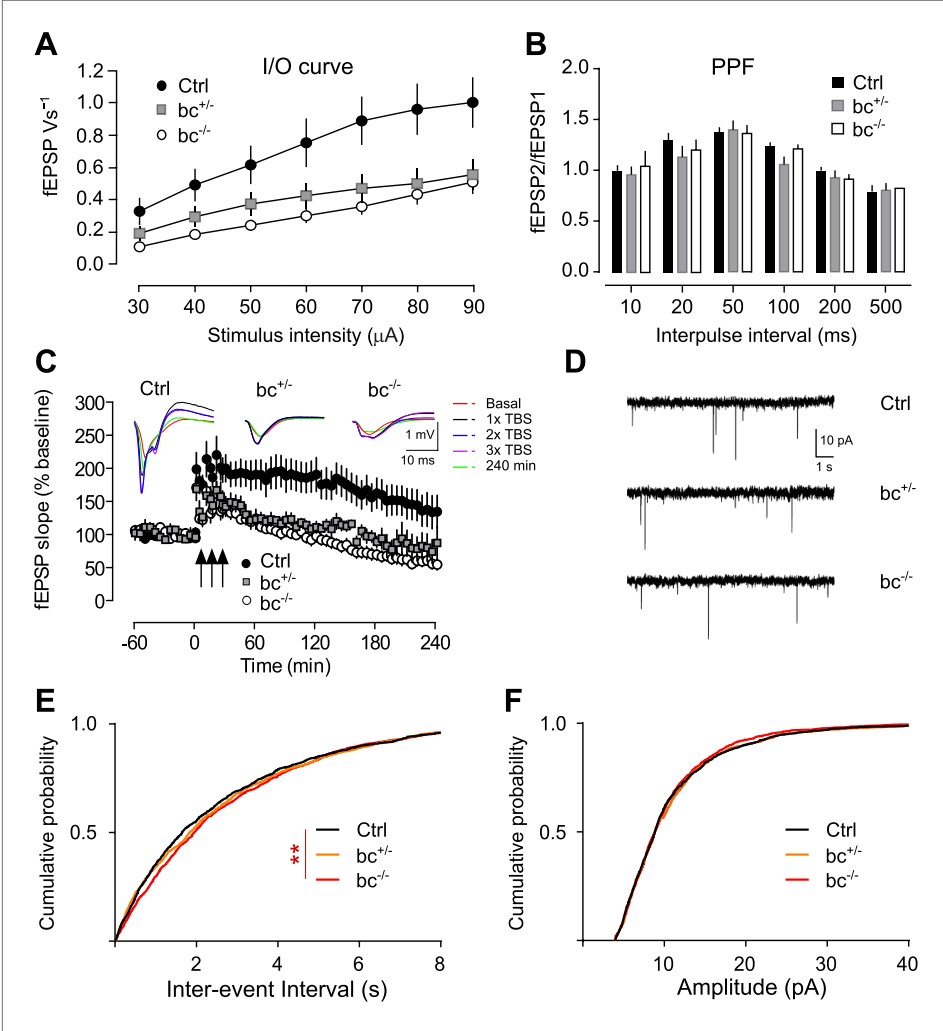

**Figure 2**. *Aph1bc* deletion impairs synaptic transmission and plasticity. (**A**) Input-Output curves recorded in the Schaffer collaterals of the hippocampus show that basic synaptic transmission is impaired in *Aph1bc*[+/−] heterozygous and homozygous *Aph1bc*[−/−] mutant mice as compared to control littermates. Graph shows means ± SEM, RM-ANOVA for the three groups: $F_{(2,18)} = 4.163$, $p<0.05$; Ctrl: $n = 6$; *Aph1bc*[+/−]: $n = 7$; *Aph1bc*[−/−]: $n = 6$. (**B**) Paired pulse facilitation (PPF), a presynaptic form of short term synaptic plasticity, is not significantly affected by genetic *Aph1bc* deletion. RM-ANOVA, $p>0.05$. (**C**) Long term potentiation elicited by three bursts of theta stimulations (black arrows) is reduced in heterozygous and homozygous *Aph1bc* mutant mice in comparison to control mice. The insets show representative traces from mutant and control mice. Means ± SEM, RM-ANOVA for the three groups. $F_{(2,18)} = 9.74$, $p=0.0014$. Ctrl $n = 6$; *Aph1bc*[+/−] $n = 7$; *Aph1bc*[−/−] $n = 6$. (**D**) Representative traces from mEPSC recordings in slices from control, *Aph1bc*[+/−] and *Aph1bc*[−/−] mice plotted in (**E**) and (**F**). (**E**) Cumulative plot of inter-event intervals of mEPSCs in control, heterozygous and homozygous *Aph1bc* deficient mice. Kruskal–Wallis test followed by Dunn's multiple comparison test, **$p<0.01$. Median, Ctrl 1651 ms , BC[+/−] 1835 ms, BC[−/−] 1947 ms. Mean, Ctrl 2465 ms ± 56, BC[+/−] 2550 ms ± 58, BC[−/−] 2622 ms ± 56. $n >1679$ each out of 37 control, 36 *Aph1bc*[+/−] and 35 *Aph1bc*[−/−] neurons. (**F**) Cumulative probability of mEPSCs amplitude in control and mutant *Aph1bc* mice. Komolgorov–Smirnov test, $p>0.05$.

conditional mutant mice. We transfected *Aph1bc*[fl/fl] neurons at DIV8 with GFP as a control or with GFP-*ires*-Cre to obtain single-cell deletion of Aph1bc, and we quantified dendritic spine density at DIV15. This study showed that Aph1bc loss of function impaired the formation of dendritic spines (*Figure 3A–C*). We then hypothesized that a deficit in Nrg1 intracellular signalling underpinned the Aph1bc loss of function phenotype. This model predicts that restoring Nrg1 intracellular signalling would rescue the observed dendritic spine deficit in Aph1bc[−/−] neurons. Therefore,

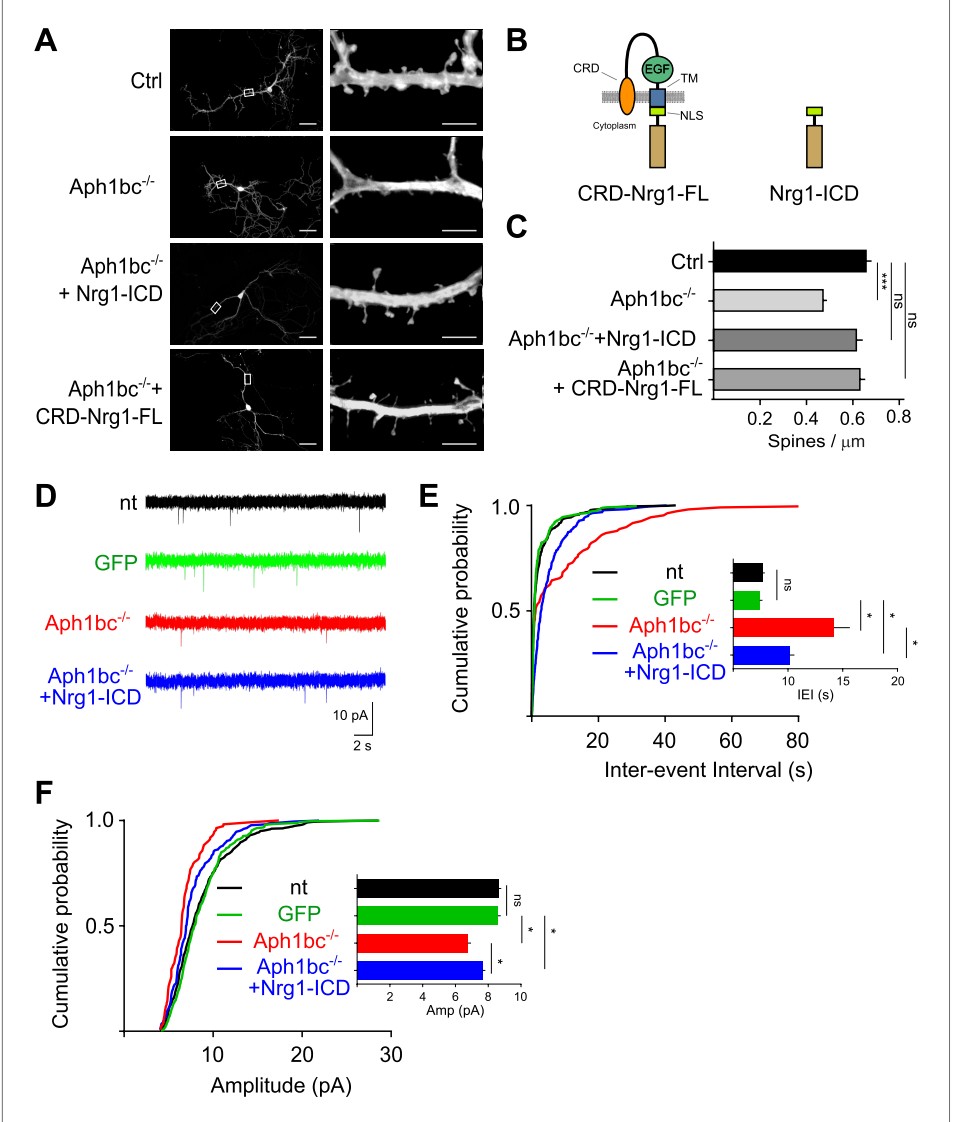

**Figure 3**. Spine formation is impaired by single cell Aph1bc-γ-secretase loss of function and is rescued by Nrg1 intracellular signalling. (**A**) Representative pictures at DIV15 of cultured hippocampal from *Aph1bc*<sup>fl/fl</sup> conditional mutant mice transfected at DIV8 with GFP as control, GFP-*ires*-Cre to obtain single cell *Aph1bc*<sup>−/−</sup> neurons, GFP-*ires*-Cre and CRD-Nrg1-FL or Nrg1-ICD to restore Nrg1 intracellular signalling in *Aph1bc*<sup>−/−</sup> neurons. Scale bars in left column: 50 μm, right column: 5 μm. (**B**) The schemata show the structure of CRD-Nrg1 full length (CRD-Nrg1-FL), of Nrg1 intracellular domain (Nrg1-ICD). CRD, Cysteine Rich Domain, EGF epidermal growth factor-like domain; TM, transmembrane domain; NLS, nuclear localization signal. (**C**) Quantification of spine density. Selective single cell genetic deletion of Aph1bc-γ-secretase decreased spine density. Co-expression of CRD-Nrg1-FL and of Nrg1-ICD in *Aph1bc*<sup>−/−</sup> neurons rescued the impairment in spine formation. Means ± SEM, one-way ANOVA. ***p<0.001. Ctrl: n = 36; *Aph1bc*<sup>−/−</sup>: n = 36; *Aph1bc*<sup>−/−</sup>+CRD-Nrg1-FL: n = 38; *Aph1bc*<sup>−/−</sup>+Nrg1-ICD: n = 37. (**D**) Representative traces from mEPSC recordings shown in (**E**) and (**F**). (**E**) Cumulative probability of inter-event intervals of mEPSCs recorded in non-transfected and GFP positive control neurons, in *Aph1bc*<sup>−/−</sup> deficient neurons and in *Aph1bc*<sup>−/−</sup> neurons transfected with Nrg1-ICD. nt = non transfected. The inset graph shows means ± SEM. Kruskal–Wallis test followed by Dunn's multiple comparison test, ns p>0.05; *p<0.05. nt: n = 457 out of 9 neurons; GFP: n = 359 out of 6 neurons; *Aph1bc*<sup>−/−</sup>: n = 107 out of 7 neurons; *Aph1bc*<sup>−/−</sup>+Nrg1-ICD: n = 220 out of 8 neurons. (**F**) Cumulative probability plot of mEPSCs amplitude recorded in non-transfected or GFP positive control neurons, in *Aph1bc*<sup>−/−</sup> deficient neurons and in *Aph1bc*<sup>−/−</sup> neurons transfected with Nrg1-ICD. nt = non transfected. The inset graph shows means ± SEM, Kruskal–Wallis test followed by Dunn's multiple comparison test, ns, p>0.05; *p<0.05.

we co-transfected GFP-*ires*-Cre with Nrg1-ICD or with CRD-Nrg1-FL (*Figure 3B*). The expression of Nrg1-ICD indeed rescued the impairment of spine formation in *Aph1bc*$^{-/-}$ neurons in a cell autonomous way (*Figure 3A–C*). Moreover, also CRD-Nrg1-FL expression rescued the Aph1bc dependent phenotype (*Figure 3A–C*).

To further asses the cell autonomous relevance of Aph1bc-γ-secretase/Neuregulin 1 intracellular signalling in excitatory synaptic function, we recorded mEPSCs in neuronal cultures. As additional control we also measured mEPSCs in non-transfected (nt) neurons which were undistinguishable from GFP expressing neurons. The IEIs and the amplitude of mEPSCs were impaired in *Aph1bc*$^{-/-}$ relative to control neurons (*Figure 3D–F*). This phenotype was substantially, although not completely, rescued by the expression of Nrg1-ICD in *Aph1bc*$^{-/-}$ neurons (*Figure 3D–F*). These results are coherent with the morphological analysis of dendritic spine density and further support a role for Aph1bc-γ-secretase/Nrg1 intracellular signalling in excitatory connections.

## Selective function of different γ-secretase complexes in spine formation

Pyramidal neurons express both Aph1a- and Aph1bc-γ-secretase complexes (*Serneels et al., 2009*). We reasoned that the rescue of spine formation observed upon exogenous expression of CRD-Nrg1-FL might indicate that Aph1a-γ-secretase, which is also expressed by pyramidal neurons (*Serneels et al., 2009*), could compensate for the loss of Aph1bc-dependent Nrg1 processing under these experimental conditions of overexpression of the substrate. Hence, we investigated the involvement of Aph1a in dendritic spine development in two additional experimental paradigms. First, we analysed hippocampal cultures from *Aph1a*$^{fl/fl}$ conditional mutant mice. Aph1a deletion by GFP-*ires*-Cre expression did not alter spine formation as compared to control neurons expressing GFP, suggesting that Aph1a has a redundant function in spine formation in vitro when Aph1bc is present (*Figure 4A,B*).

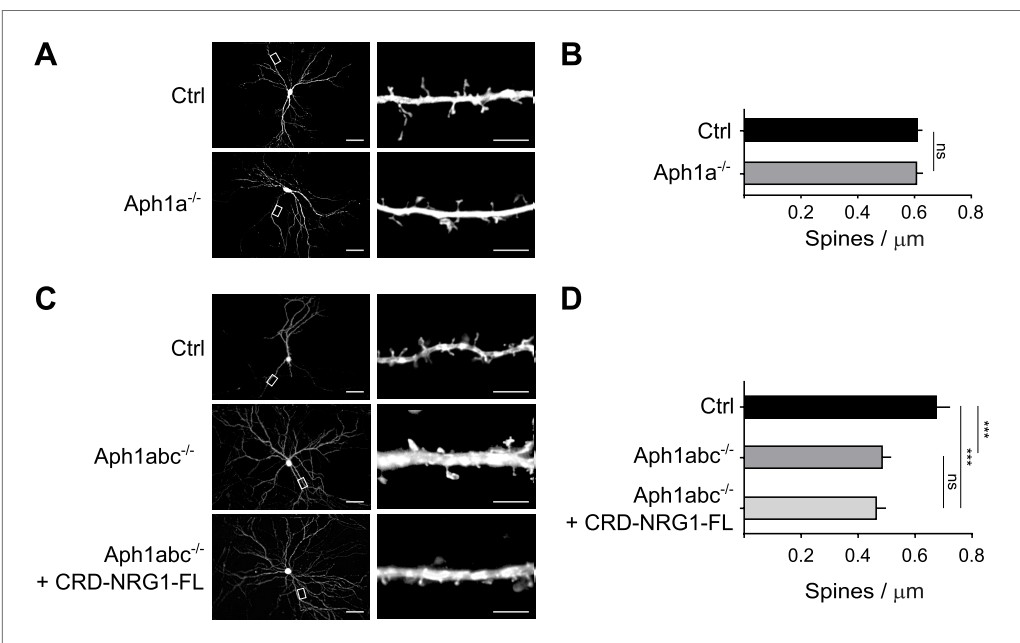

**Figure 4**. Selective function of different γ-secretase complexes in spine formation. (**A**) Cultured hippocampal neurons from *Aph1a*$^{fl/fl}$ conditional mutant mice were transfected at DIV8 with GFP as control or with GFP-*ires*-Cre to delete *Aph1a* and fixed at DIV15. (**B**) Single cell deletion of *Aph1a* indicates that Aph1a-γ-secretase activity is not necessary for spine formation in these experimental conditions. Means ± SEM, *t* test. p>0.05. Ctrl: n = 25; *Aph1a*$^{-/-}$: n = 19. (**C**) Hippocampal neurons from *Aph1abc*$^{fl/fl}$ triple conditional mutant mice were transfected with GFP as control, with GFP-*ires*-Cre to completely abrogate γ-secretase activity in single neurons or co-transfected with GFP-*ires*-Cre and CRD-Nrg1-FL. (**D**) Complete γ-secretase loss of function by *Aph1abc*$^{-/-}$ triple deletion impaired spine formation. This phenotype could not be rescued by CRD-Nrg1-FL indicating that γ-secretase dependent Nrg1 intracellular signalling is necessary to restore spine formation. Means ± SEM, one-way ANOVA. ***p<0.001. Ctrl: n = 13; *Aph1abc*$^{-/-}$: n = 22; *Aph1bc*$^{-/-}$ + CRD-Nrg1-FL: n = 19. Scale bars in **A**, **C**, left column: 50 μm, right column: 5 μm.

We then established primary neuronal cultures from *Aph1abc*[fl/fl] triple conditional mutant mice to obtain complete genetic abrogation of γ-secretase activity (*Serneels et al., 2009*). Deletion of all of the *Aph1* genes by GFP-*ires*-Cre expression at DIV8 decreased spine density at DIV15 similarly to Aph1bc deletion. Moreover, expression of CRD-Nrg1-FL did not rescue decreased spine density in *Aph1abc*[-/-] triple mutant neurons, indicating that CRD-Nrg1-FL cleavage by the γ-secretase is necessary to rescue spine formation (*Figure 4C,D*). Taken together with the conditional Aph1bc loss of function experiments, these data provide a proof of concept that specific γ-secretase complexes are differentially involved in spine formation. In addition, they indicate that γ-secretase is required to trigger Nrg1 intracellular signalling in this biological process.

## Nrg1 intracellular signalling increases dendritic spine formation

We further established the role of Nrg1 in spine formation by performing additional gain-of-function experiments. We transfected wild type hippocampal neurons in vitro with GFP as control or with constructs co-expressing GFP and Nrg1 at DIV8, and we analysed transfected neurons at DIV15. We found that expression of CRD-Nrg1-FL increased spine density as compared to control neurons expressing GFP only (*Figure 5A,B*). To selectively test the role of Nrg1 intracellular signalling, we co-transfected neurons with a construct encoding Nrg1-ICD. Exogenous expression of Nrg1-ICD increased the density of dendritic spines as compared to controls, indicating that the activation of Nrg1 intracellular signalling promotes spine formation (*Figure 5A,B*). Nrg1-ICD contains a nuclear localization signal (NLS), which is required for the translocation of Nrg1-ICD to the nucleus where it regulates gene expression (*Bao et al., 2003*). Here, we found that the expression of the Nrg1-ICD lacking the NLS (Nrg1-ΔNLS-ICD) does not alter the formation of dendritic spines (*Figure 5A,B*). Collectively, these data indicate that cell autonomous Nrg1 intracellular signalling cell autonomously enhances dendritic spine formation and the localization of the Nrg1-ICD to the nucleus is required for this function.

## Aph1bc loss-of-function impairs spine formation which is rescued by Nrg1-ICD in vivo

The observation that *Aph1bc* was required for spine formation in neuronal cultures in vitro led us to investigate the neuronal morphology and spine density in the brain of *Aph1bc*[−/−] deficient mice in vivo. To this aim, we compared Golgi stained neurons from *Aph1bc*[−/−] mutant and *Aph1bc*[+/+] control mice at P30. Since synaptic transmission is impaired in hippocampal Schaffer collaterals of *Aph1bc*[−/−] mice, we scrutinized spine density in dendrites of CA1 hippocampal neurons that receive input from CA3

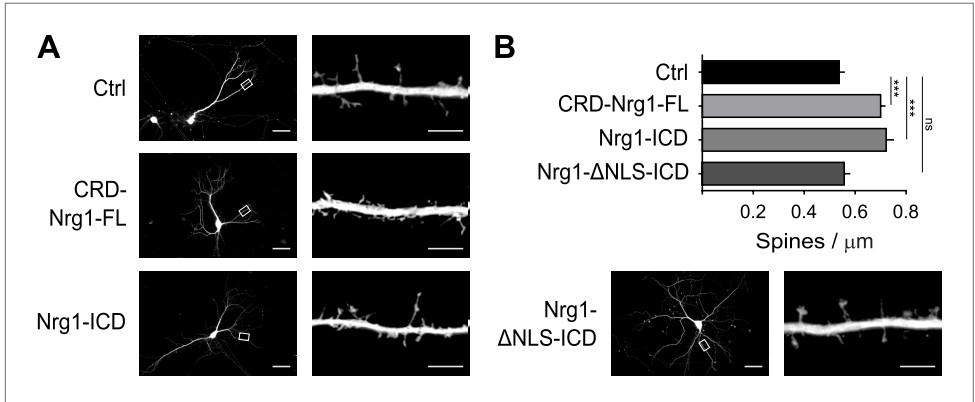

**Figure 5**. Nrg1 intracellular signalling cell-autonomously promotes spine formation in vitro. (**A**) Representative pictures of cultured hippocampal neurons transfected at DIV8, at the beginning of synaptogenesis, with either GFP alone as control or GFP and the CRD-Nrg1-Fl, GFP and the Nrg1-ICD and GFP and Nrg1-ΔNLS-ICD. Neurons were fixed and analysed at DIV15. (**B**) Quantification of spine density in Nrg1 transfected neurons. Single cell exogenous expression of Nrg1-Fl and of Nrg1-ICD enhanced spine formation. Conversely, Nrg1-ΔNLS expression did not increase spine density indicating that nuclear localization signal of Nrg1 is required for this function. Means ± SEM, one-way ANOVA. ***p<0.001. Ctrl, n = 19; CRD-Nrg1-FL, n = 15; Nrg1-ICD, n = 16; Nrg1-ΔNLS-ICD, n = 21. Scale bars in **B**, left column: 50 μm, right column: 5 μm.

axons. Morphological observation of *Aph1bc⁻/⁻* neurons did not reveal overt abnormalities (*Figure 6A*). In addition, Sholl analysis did not show a significant difference in dendritic arborisation between *Aph1bc⁺/⁺* control and *Aph1bc⁻/⁻* mice (*Figure 6B*). However, we found that dendritic spine density was significantly decreased in the apical dendrites of *Aph1bc⁻/⁻* neurons as compared to controls (*Figure 6C,D*).

Our in vitro experiments suggest that the role of *Aph1bc* was cell autonomous and that expression of the Nrg1-ICD could rescue Aph1bc loss of function. Thus, to investigate the cell autonomous role of Aph1bc under physiological conditions, we analysed the effect of Aph1bc deletion in single cells developing on a wild type background. For that we performed in utero electroporation (IUE) of GFP or GFP-*ires*-Cre in *Aph1bc^fl/fl* conditional mice. We found that single cell conditional deletion of Aph1bc

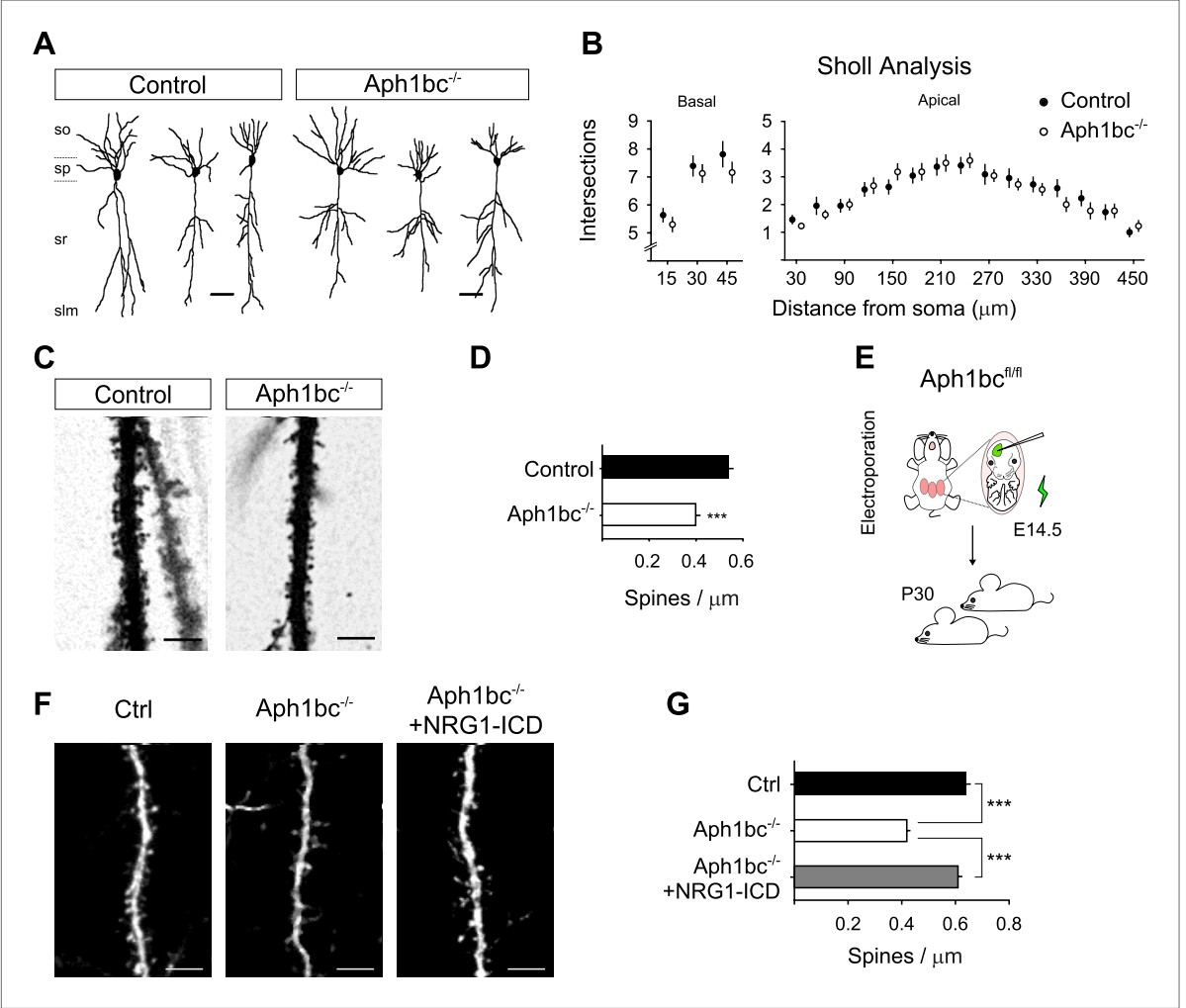

**Figure 6**. *Aph1bc* deletion cell autonomously disrupts spine formation which is rescued by Nrg1-ICD expression in vivo. (**A**) Representative drawings of Golgi stained CA1 hippocampal neurons from control and *Aph1bc⁻/⁻* null brains at P30. so, stratum oriens; sp, stratum pyramidale; sr, stratum radiatum; slm, stratum lacunosum moleculare. (**B**) Sholl analysis of dendritic arbour of neurons from *Aph1bc⁻/⁻* mice did not reveal overt defects in neuronal morphology as compared to control in neither basal nor apical dendrites. Basal, means ± SEM, two-way ANOVA. p>0.05. Ctrl: n = 40; *Aph1bc⁻/⁻*: n = 32. Apical; Apical, means ± SEM, two-way ANOVA. p>0.05. Ctrl: n = 22; *Aph1bc⁻/⁻*: n = 22. (**C**) Representative images of apical dendrites of CA1 hippocampal neurons that receive input from Schaffer collaterals from control and *Aph1bc⁻/⁻* mice. (**D**) Histogram shows that spine density is decreased in apical dendrites of *Aph1bc⁻/⁻* deficient neurons. Means ± SEM, *t* test. p<0.001. Ctrl, n = 31; *Aph1bc⁻/⁻*, n = 46. (**E**) Schema summarizing the experimental paradigm for cell autonomous Aph1bc loss of function and rescue by Nrg1-ICD via in utero electroporation (IUE) at E14.5. (**F**) Basal dendrites of layer II/III cortical pyramidal neurons from *Aph1bc^fl/fl* mutant mice electroporated at E14.5 with either GFP as control, GFP-*ires*-Cre alone to perform single cell *Aph1bc⁻/⁻* deletion or with GFP-*ires*-Cre and Nrg1-ICD to rescue spine formation and fixed at P30. (**G**) Quantification of spine density. Spine formation was impaired by single cell deletion of *Aph1bc* and it was rescued by expression of Nrg1-ICD construct in *Aph1bc⁻/⁻* neurons. Means ± SEM. One-way ANOVA. ***p<0.001. Ctrl, n = 41; *Aph1bc⁻/⁻*, n = 78; *Aph1bc⁻/⁻*+Nrg1-ICD, n = 47. Scale bar in **A** 50 μm, in **C** and **F** 5 μm.

impaired spine formation as compared to GFP electroporated neurons (*Figure 6E–G*). Moreover, co-electroporation of Nrg1-ICD together with GFP-*ires*-Cre could rescue decreased spine density (*Figure 6E–G*). Altogether, these data indicated that Aph1bc-γ-secretase activity cell autonomously controls dendritic spine formation in vivo at least in part via the activation of Nrg1 intracellular signalling.

## Discussion

Here we scrutinized the involvement of *Aph1bc* in neuronal synaptic function and investigated the hypothesis that Nrg1 intracellular signalling controls synaptogenesis downstream of Aph1bc-γ-secretase. *Aph1bc⁻/⁻* deficient brains did not show overt signs of neuronal degeneration or an alteration in neuronal layering or dendritic branching. Nonetheless, we found that the expression of excitatory synaptic markers, synaptic transmission, mEPSCs and long-term plasticity were impaired in *Aph1bc⁻/⁻* deficient mice. Taking advantage of conditional selective γ-secretase mutant mice, we demonstrated in neuronal cultures that Aph1bc is cell autonomously required for dendritic spine formation and that Nrg1-ICD and CRD-Nrg1-FL expression could rescue the *Aph1bc* loss of function phenotype. The relevance of Aph1bc-γ-secretase/Nrg1 intracellular signalling in excitatory synaptic function was further supported by the impairment in mEPSCs in *Aph1bc⁻/⁻* neuronal cultures which was significantly alleviated by Nrg1-ICD expression. Notably, exogenous expression of CRD-Nrg1-FL could not rescue the spine formation deficit in *Aph1abc⁻/⁻* neurons, which are completely devoid of γ-secretase activity (*Serneels et al., 2005*). Therefore, the γ-secretase-mediated cleavage of Nrg1 is necessary for Nrg1 rescue of spine formation. Besides, even though Aph1a-γ-secretase might redundantly contribute to Nrg1 processing, collectively our data indicate that Aph1bc-γ-secretase is the major regulator of Nrg1 intracellular signalling in this biological process. Finally, single cell Aph1bc deletion in vivo by in utero electroporation impaired spine formation and could be rescued by Nrg1-ICD expression, emphasizing the physiological relevance of our observations.

Our study indicates that the Aph1bc-γ-secretase complex controls the establishment of excitatory circuits under physiological conditions at least in part through the regulation of Nrg1 intracellular signalling. It should be kept in mind that both γ-secretase complexes and Nrg1 isoforms impinge on multiple signalling molecules that control many facets of neuronal development and function. As several putative γ-secretase substrates have been identified it is still possible that some of them may also contribute to the behavioural deficits of *Aph1bc⁻/⁻* mice (*Dejaegere et al., 2008*). Nonetheless, we have previously shown that deletion of Aph1bc does not affect the processing of other major γ-secretase substrates such as Notch and ErbB4 in vivo (*Serneels et al., 2005*; *Dejaegere et al., 2008*). In addition, mutations of other γ-secretase substrates such as APP or N-Cadherin have not been previously shown to cause schizophrenia-like behavioural deficits in mice or to be associated with schizophrenia. Hence, the biochemical and cell biological evidences from the current work, and the available genetic evidence are consistent with the conclusion that Nrg1 is the major physiological target of Aph1bc-γ-secretase in vivo in the context of synapse formation. The pathological relevance of human *APH1B* and *NRG1* genetic interaction and of NRG1 processing is further supported by recent studies in Schizophrenic patients linking polymorphisms in these genes to the disease (*Marballi et al., 2012*; *Hatzimanolis et al., 2013*).

Nrg1 forward signalling controls the establishment of inhibitory circuits in the cortex by activating its specific receptor ErbB4, which is primarily expressed in cortical interneurons (*Fazzari et al., 2010*; *Wen et al., 2010*). On the other hand, the physiological role of CRD-Nrg1 intracellular signalling via the Nrg1 intracellular domain could not be addressed unambiguously in vivo in available mutant mice since deletion of CRD or TM domain of Nrg1 protein would affect both forward and intracellular signalling.

Previous in vitro studies proposed that γ-secretase dependent Nrg1 signalling may control the expression of genes that control neuronal survival (*Bao et al., 2003*) and dendritic growth during development (*Chen et al., 2010a*). Our in vivo experiments using Aph1bc-γ-secretase deficient brains do not demonstrate neuronal loss or impaired dendritic arborisation, similar to the observations in Nrg1 transmembrane domain mutant heterozygous mice (*Zhang et al., 2009*). We speculate that these differences may be explained by differential effects of long term abrogation of Nrg1 intracellular signalling in Nrg1 constitutive null mice which is not the case in *Aph1bc⁻/⁻* conditional mice. On the other hand, our results show that Aph1bc-dependent Nrg1 intracellular signalling promotes dendritic spine formation. Consistent with these findings, schizophrenia-like

deficits and impaired maturation of glutamatergic synapses have also been described for mice deficient in Nrg1, ErbB4 and BACE1, a protease that initiates Nrg1 processing by cleaving its extracellular domain (*Chen et al., 2008*; *Barros et al., 2009*; *Del Pino et al., 2013*). Although these deficits were attributed to loss of ErbB4 activation in pyramidal neurons, this interpretation was challenged by the recent discovery that ErbB4 is almost exclusively expressed by cortical interneurons (*Fazzari et al., 2010*; *Wen et al., 2010*). More recently, it was proposed that altered glutamatergic wiring could be the result of a homeostatic response to alterations in ErbB4 expressing fast-spiking interneurons (*Del Pino et al., 2013*). Here we propose a complementary, but not mutually exclusive, mechanistic model whereby dendritic spine maturation could directly, in a cell-autonomous manner, be promoted by Aph1bc-dependent Nrg1 intracellular signalling. Notably, it has been suggested that Nrg1 could be involved in activity-dependent regulation of the structural plasticity of glutamatergic circuits since neuronal activity enhances both Nrg1 expression (*Eilam et al., 1998*) and proteolytic processing by γ-secretase (*Bao et al., 2003*; *Ozaki et al., 2004*). Consistently, we show here that Aph1bc-γ-secretase-dependent Nrg1 intracellular signalling promotes spine formation.

In conclusion, we provide a cellular and molecular mechanism for the cognitive deficits observed in Aph1bc-γ-secretase-deficient mice. Moreover, our study suggests that schizophrenia linked cSNPs in TM domain of NRG1 (*Walss-Bass et al., 2006*; *Mei and Xiong, 2008*) and mutations in *APH1B* gene (*Hatzimanolis et al., 2013*) may contribute to the alteration of dendritic spines density observed in schizophrenia (*Lewis and Sweet, 2009*; *Glausier and Lewis, 2013*), linking APH1B and NRG1 misprocessing firmly to this disorder.

## Materials and methods

### Generation of mice

*Aph1a*$^{fl/fl}$ and *Aph1bc*$^{fl/fl}$ were previously described (*Serneels et al., 2005*). To obtain Aph1bc deficient brains for WB and IF analysis *Aph1bc*$^{fl/fl}$ mice were crossed with heterozygous Nestin driven Cre mice (B6.Cg-Tg(Nes-cre)$^{1Kln/J}$; Jackson Laboratory, Bar Harbor, ME), Cre negative littermates were taken as controls. To obtain homozygous and heterozygous *Aph1bc* null mice for electrophysiology, *Aph1bc*$^{fl/fl}$ were first crossed with *Pgk* driven Cre as described (*Serneels et al., 2005*) to obtain constitutive *Aph1bc*$^{-/-}$ that were backcrossed with wild type mice. Neuronal cultures were performed from conditional *Aph1bc*$^{fl/fl}$ embryos. All colonies were kept in C57BL/6J background and littermates were taken as controls. All experiments were approved by the Ethical Committee on Animal Experimenting of the University of Leuven (KU Leuven).

### Immunolabelling, Golgi staining, imaging and analyses

Mice were transcardially perfused with PBS followed by freshly prepared 4% PFA in PBS, cut with cryostat at 40 μm, and eventually processed for immunofluorescence on floating sections as described (*Fazzari et al., 2010*) or cut with Vibratome at 100 μm and processed with FD Rapid Golgistain kit (PK401; FD NeuroTechnologies) according to manufacturer instructions. Antibodies: mouse anti-NeuN (MAB377; 1:500; Chemicon-Millipore); rabbit anti-CDP (Cux1) (sc-13024; 1:200; Santa Cruz Biotechnology); mouse anti-VGluT1 (MAB5502; 1:200 in IF; Chemicon); rabbit anti-Homer1 (160003; 1:500; synaptic systems); rabbit anti-VGAT (131003; 1:500; synaptic systems); mouse anti-Parvalbumin (235; 1:500; Swant) chicken anti-GFP (GFP-1020; 1:1000; Aves). All secondary antibodies were conjugated with Alexa Fluor® dyes (Life technologies). Conventional imaging was performed with a Zeiss Axioplan2 upright microscope with 20x Plan-Apochromat (NA = 0.5) and 100x Plan-Apochromat oil immersion (NA = 1.4) objectives. For confocal imaging we used Olympus FV1000 IX2 Inverted Confocal microscope with 60x UPlanSapo (NA = 1.35) oil immersion objective. For image analyses all pictures were processed and quantified with ImageJ software. For neuronal distribution, cortices were divided in 10 bins and NeuN fluorescence intensities in each bin were normalized for the total NeuN intensity. For VGluT1, Homer1, VGAT and PV puncta quantification confocal pictures (18 confocal planes out of three animals per condition) were taken 5 μm beneath tissue surface as described (*Fazzari et al., 2010*; *Iijima et al., 2011*). Images received automatically thresholds with ImageJ algorithm (Yen for VGluT1 and Homer1; Intermodes for VGAT and PV) and resulting masks were redirected to the original image for automatic quantification of puncta intensity. For PV puncta we also applied a circularity filter 0.5–1.00 and a high size cut-off filter of 1.5 μm$^2$ as previously

described (*Fazzari et al., 2010*; *Del Pino et al., 2013*). For spine quantification in neuronal cultures we took image stacks of transfected neurons (z = 0.5 µm) and we counted spines in dendritic segments at 90 to 100 µm from the soma. Morphology of Golgi stained neurons was reconstructed with ImageJ Simple neurite tracer. Spine quantification was carried out in image stacks (z = 0.5 µm) in apical dendrites of CA1 pyramidal neurons at 100 µm from pyramidal layer. In electroporated mice, we selected neurons from layer II/III and we quantified spine density in confocal stacks of basal dendrites starting from 5 µm away from the first branching point. All statistics were performed with Graph Pad Prism software.

## Western blot

Mice were sacrificed by cervical dislocation; brains were dissected in ice cold PBS and snap frozen in liquid nitrogen. Prefrontal cortices were homogenized in ice cold HEPES buffer (320 mM Sucrose; 4 mM HEPES, pH 7.3; EDTA with complete protease inhibitors from Roche); cleared by centrifugation at 800×$g$ for 10 min; equal amount of protein was loaded on NuPAGE 4–12% Bis-Tris gel (Novex, NP0322) and blotted detected with HRP conjugated secondary antibodies using a ECL chemiluminescence detection kit (NEL105; PerkinElmer Life Sciences). All antibodies were diluted in 1% BSA TBST buffer. Antibodies: mouse anti-VGluT1 (MAB5502; 1:2000; Chemicon) mouse anti-Synaptophysin (S5768; 1:5000; Sigma); mouse anti-PSD95 (ADI-VAM-PS002-E; 1:5000; Stressgen); rabbit Anti-Tubulin (ab21058; 1:5000; abcam). The density of bands was quantified by densitometry using ImageJ software.

## Electrophysiology

For field recordings hippocampal slices were prepared as described (*Denayer et al., 2008*). Briefly, 6–8 weeks old mice, were killed by cervical dislocation and the hippocampus was rapidly dissected into ice-cold artificial cerebrospinal fluid (ACSF, pH 7.4, saturated with carbogen, 95% O2/5% CO2). Transverse slices (400 µm thick) were prepared from the dorsal area and placed into a submerged-type slice chamber, where they were maintained at 33°C and continuously perfused with carbogen-saturated ACSF. After 90 min incubation, tungsten stimulating electrodes and glass recording electrodes were placed into the stratum radiatum of hippocampal area-CA1 to evoke field excitatory post-synaptic potentials (fEPSP). To assess basic properties of synaptic responses, I/O curves were established by stimulation with 30–90 µA constant currents (pulse width 0.1 ms). The stimulation strength was adjusted to evoke a fEPSP-slope of 35% of the maximum and kept constant throughout the experiment. Paired pulse facilitation (PPF) was examined by applying two pulses in rapid succession (interpulse intervals of 10, 20, 50, 100, 200, and 500 ms, respectively) at 120 s intervals. 60 min thereafter, baseline recordings were started consisting of three single stimuli with 0.1 ms pulse width repeated at a 10-s interval and averaged every 5 min. LTP was induced by three TBS episodes separated by 10 min, with evoked responses monitored at 1, 4 and 7 min between TBS episodes. 10 min after the last TBS episode, evoked responses were recorded every 5 min during 4 hr. In all experiments, the recording of slices from mutant mice was interleaved by experiments with wild type controls.

Patch-clamp recordings of mEPSCs were performed in acute hippocampal slices obtained from control and *Aph1bc* mutant siblings. Animals were decapitated and the brain was quickly removed and placed in an ice-cold artificial cerebrospinal fluid (ACSF) containing (in mM) 124 NaCl, 4.9 KCl, 1.2 NaH2PO4, 25.6 NaHCO3, 2 MgSO4, 2 CaCl2, and 10 glucose, and saturated with 95% O2 and 5% CO2 (pH 7.3–7.4). Transverse hippocampal slices (400 µm thick) were cut with a vibratome (HM 650V; 'MIKROM') and stored at room temperature in a holding bath (pre-chamber) containing the same ACSF as above. After a recovery period of at least 1 hr, an individual slice was transferred to the recording chamber where it was continuously superfused with oxygenated ACSF at a rate of 2.5 ml/min.

Pyramidal neurons in the CA1 region of the hippocampus were visually identified using infrared-differential interference contrast (DIC) microscopy. Whole-cell recordings were obtained using a patch-clamp amplifier (MultyClamp 700B; Axon Instruments). Patch pipettes (resistance 3–5 MΩ) were pulled from borosilicate glass using a horizontal puller (Model P-97; Sutter Instruments, Novato, CA) and were filled with a solution containing: 135 mM CsMeSO4, 4 mM NaCl, 4 mM MgATP, 0, 3 mM Na-GTP, 0,5 mM EGTA, 10 mM K-HEPES; pH 7.24; 281 mOsm.

Voltage-clamp recordings of mEPSCs were performed in ACSF supplemented with 1 µM tetrodotoxin (TTX) and 100 µM picrotoxin (PicTX). Holding voltage −70 mV. Data were low-pass filtered at 2 kHz and acquired at 10 kHz using Digidata 1440 and pClamp 10 software. Off-line analysis of mEPSCs was performed using MiniAnalysis (v.6.0.7, Synaptosoft, Decatur, GA) software.

Whole-cell recordings in neuronal cultures were performed at DIV20 similarly as described above. Briefly, cells were visualized on a Nikon Eclipse FN1 microscope with a Hamamatsu C10-600 camera; mEPSCs recordings were performed in low fluorescence Hibernate E neuronal culture medium (HE-If, Brainbits UK) supplemented with 1 μM tetrodotoxin (TTX). Holding voltage −70 mV.

## Neuronal cultures

Hippocampi from E17.5–18.5 mouse embryos were dissected, treated with trypsin and dissociated into single cells by gentle trituration. Cells were resuspended in MEM (cat no 31095029; Invitrogen) containing 10% Horse serum (cat no 26050088; Invitrogen), penicillin–streptomycin and 0.6% glucose, and then plated at a density of 1000 cells per mm$^2$ on coverslips coated with 1 mg ml$^{-1}$ poly-L-lysine (P2636-1g; Sigma) and laminin 5 μg ml$^{-1}$ (cat no 3400-010-01; R&D systems). After 2 hr, the plating medium was replaced by Neurobasal medium (Invitrogen), penicillin–streptomycin and B27 supplements (cat no 17504-044; Invitrogen) as described (*Fazzari et al., 2010*). Transfection was performed with Lipofectamine 2000 (Invitrogen) according to manufacturer instructions. Briefly, neuronal culture medium was taken and replaced with Lipofectamine/DNA mix diluted in Neurobasal medium that was left on neuronal culture for 90–120 min; next, saved conditioned medium was put back on neuronal cultures. After DIV10, half of the neuronal culture medium was refreshed every other day. Plasmids: pCMV-GFP (Clontech), pCMV-GFP-*ires*-Cre (*Fazzari et al., 2010*); CRD-Nrg1 full length tagged with GFP was kindly provided by Prof. Bao Jianxin (*Bao et al., 2003*), and cloned into pcDNA3.1 TOPO (Invitrogen) with forward primers 5'-AAA TAA GGC GCC ACT ATA GGG AGA CCC AAG CTG GC-3', 5'-AAA TAA GGC GCC ATG AAA ACC AAG AAA CAG CGG CAG AAG C-3' and 5'-AAA TAA GGC GCC ATG CAG AGC TTC GGT CA GAA CGA AAC-3' for CRD-Nrg1-FL, Nrg1-ICD and Nrg1-ΔNLS-ICD respectively, adapted from *Bao et al. (2003)* and reverse primer 5'-AAT AAT GTC GAC CAA ACA ACA GAT GGC TGG CAA CTA GAA G-3' for all constructs. The correct expression of all plasmids was tested by WB (not shown). Immunofluorescence was performed as described (*Fazzari et al., 2010*).

## In utero electroporation

In utero electroporation was performed as described (*Shariati et al., 2013*). Pregnant mice were anaesthetized by intramuscular injections of 88 mg ketamine and 132 mg xylazine per gram of body weight. The uterine horns were exposed and the plasmids mixed with Fast Green (Sigma) were micro-injected in the lateral ventricles of E14.5 embryos. Five current pulses (50 milliseconds pulse/950 milliseconds interval) were delivered across the head of the embryos (36 V) targeting the dorsal-medial part of the cortex. An equal amount of pCAG-ires-GFP (0.5 μg/μl) was electroporated in all conditions to ensure an equal visualization of neuronal morphology. Plamids: pCAG-ires-GFP (Add Gene, 11,159); pCMV-GFP-*ires*-Cre (*Fazzari et al., 2010*); Nrg1-ICD was subcloned from pcDNA3.1 (see above) to pCAGEN (11160; Add Gene). All animal experiments were approved by the Ethics Committee of the KU Leuven.

## Acknowledgements

We thank Prof. Bao Jianxin for kindly providing the construct for CRD-Nrg1. We thank Carlos Dotti, Joris De Wit, Claudia Bagni, Amantha Thathiah and Vanessa Morais for critical discussions of the study. We thank Véronique Hendrickx and Jonas Verwaeren for technical assistance. We thank Jan Slabbaert for the help with electrophysiological recordings in neuronal cultures and Inframouse facility at K.U.Leuven for histological infrastructure. This work was supported by FWO Foundation for Scientific Research Belgium, research grant EME-C3957-G.0512.12; SAO Alzheimer's Research Foundation grant SAO-FRA P#11005; KULeuven; Federal Office for Scientific Affairs (IAP P7/16); a Methusalem grant of the Flemish Government/KU Leuven, VIB, IWT, the European Research Council (ERC); BDS is the Arthur Bax and Anna Vanluffelen chair for Alzheimer's disease.

## Additional information

### Competing interests

BDS: Reviewing editor, *eLife*, and it might be perceived as a potential conflict of interest that I (BDS) am consultant for Janssen Pharmaceutica, Remynd NV and Envivo Pharmaceutics. The other authors declare that no competing interests exist.

## Funding

| Funder | Grant reference number | Author |
| --- | --- | --- |
| European Research Council (ERC) | | Bart De Strooper |
| FWO Foundation for Scientific Research | | Pietro Fazzari, Bart De Strooper |
| SAO Alzheimer's Scientific Research | | Pietro Fazzari, An Snellinx |
| Flemish Goverment | Mathusalem grant | Bart De Strooper |

The funders had no role in study design, data collection and interpretation, or the decision to submit the work for publication.

## Author contributions

PF, Conceived the project, planned the experiments, performed the experiments and analysed the results, Wrote the manuscript, Contributed unpublished essential data or reagents; AS, Performed neuronal cultures, immunohistochemistry and immunocytochemistry, Drafting or revising the article; VS, Performed and analysed electrophysiological recordings for mEPSCs, Drafting or revising the article; TA, Performed and analysed electrophysiological recordings, Conception and design, Drafting or revising the article; LS, Generated mutant mice and supervised the colony, Conception and design, Drafting or revising the article; AG, Performed in utero electroporation, Drafting or revising the article; SAMS, Performed in utero electroporation; DB, Analysed electrophysiological recordings, Conception and design, Drafting or revising the article; BDS, Conceived and supervised the project. Wrote the manuscript., Conception and design, Drafting or revising the article

## Ethics

Animal experimentation: All the experiments involving animals in this study were approved and performed in strict accordance with the recommendations of the Ethical Committee of Katholic Univesitet Leuven (Approval Nr. p047/2012). Every effort was taken to minimize suffering of mice according to the guidelines Ethical Committee.

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
