## [Decision Letter]

Thank you for sending your work entitled “Cell autonomous regulation of cortical excitatory circuitry via Aph1B-γ-secretase/Neuregulin 1 signalling” for consideration at *eLife*. Your article has been favorably evaluated by a Senior editor and 2 reviewers, one of whom is a member of our Board of Reviewing Editors.

The Reviewing editor and the other reviewer discussed their comments before we reached this decision, and the Reviewing editor has assembled the following comments to help you prepare a revised submission.

This work elegantly and systematically describes the effect of abrogation of the gamma secretase component Aph1B in synaptic transmission, plasticity, and dendritic spine formation, and provides convincing data pointing towards the neuregulin 1 intracellular domain as a downstream regulator of these events. The authors show beautiful representative dendritic spine figures and statistical analyses appear to be well performed. Work was done both with constitutive Aph1BC knock-out mice, as well as in vitro using conditional single cell deletion. A very novel and interesting part of the work was the in utero electroporation to obtain single cell conditional deletion of Aph1BC, and thereby investigate the effects of abrogation under physiological conditions in a wild-type background. This manuscript also mainly uses mouse genetic approaches and many types of rescue experiments, and thus the data are largely convincing. Lastly, given the increasing demand for molecular mechanisms underlying schizophrenia, these findings may help us better understand the etiology of schizophrenia.

Major comments:

1) This manuscript needs to provide electrophysiological data that would strongly support their structural data on dendritic spines in Figures 3, 4 and 5. The authors do not have to try electrophysiological measurements for all the structural data, but at least some of the key results should be accompanied with functional data. For instance, they can measure mEPSCs from cultured hippocampal neurons or brain slices. Measuring mEPSCs in Figure 2 would also help.

2) The current manuscript is mainly focusing on excitatory synapses situated in dendritic spines, but lacks data for inhibitory synapses. The authors need to visualize inhibitory synaptic markers such as VGAT and gephyrin (Figure 1), and measure mIPSCs for some of the results described in Figures 3, 4 and 5.

Minor comments:

1) The Sholl analysis shown in Figure 5 seems to analyze only the proximal regions of the dendrites (∼50 microns). But the dendrites of the CA1 neurons, especially the apical dendrites, are much longer, extending several hundred microns. The coverage should be increased. In addition, the SLM layer of CA1 is not indicated in the figure.

2) Given the majority of the experiments were performed in the hippocampus, the title of the manuscript containing “…cortical excitatory circuitry…” should be toned down.

3) In the Abstract, the authors state that “…Aph1B, but not Aph1A, is selectively involved in NRG1 intracellular signaling”, but the results in Figure 3 implicate Aph1A in the processing of NRG1. This should be discussed.

4) All three panels in Figure 2 lack sample current traces.

5) The reduced LTP in Aph1BC-/- neurons (Figure 2) could be caused by the reduced basal transmission (Figure 2). This possibility should be mentioned.

6) In the Discussion, the authors appear to challenge previous work showing that impairment of neuregulin 1 intracellular formation leads to impaired dendritic arborisation (Chen et al., 2010), given that they failed to observe differences in dendritic length in the current study. However, the difference in results may be due to long-term effects of impaired NRG1 intracellular signaling, which is not suffered in the conditional deletion studies. This possibility should be acknowledged.

---

## [Author Response]

*1) This manuscript needs to provide electrophysiological data that would strongly support their structural data on dendritic spines in*
Figures 3, 4 and 5*. The authors do not have to try electrophysiological measurements for all the structural data, but at least some of the key results should be accompanied with functional data. For instance, they can measure mEPSCs from cultured hippocampal neurons or brain slices. Measuring mEPSCs in*
Figure 2
*would also help*.

As requested we performed mEPSCs recordings in acute hippocampal slices to further evaluate the impact of Aph1BC-gamma-secretase deletion on excitatory synaptic function. We observed an increase in mEPSCs inter-event intervals in Aph1BC KO mice as compared to Control mice while mEPSCs amplitude was not affected by Aph1BC deletion. These results indicate a decrease in the number of glutamate release sites that is consistent with the reduction in VGluT1 and Homer1 immunoreactivity and with the reduced spine density that we observed. In addition, we also performed recordings of mEPSCs in neuronal cultures to further corroborate the cell autonomous role of Aph1BC-gamma-secretase/Nrg1 intracellular signalling in excitatory transmission. We found that inter-event intervals and amplitude of mEPSCs are impaired by Aph1BC deletion and rescued, although not completely, by expression of Nrg1-ICD in Aph1BC deficient neurons. The experimental approach of mEPSCs recording does not allow discriminating between perisomatic, shaft, and dendritic excitatory connections, and we feel it would be too far-fetched to speculate on the reasons for this incomplete rescue. Nevertheless, these results are in line with the morphological assessment and further support the role for Aph1BC-gamma-secretase/Nrg1 intracellular signalling in excitatory synaptic function.

These novel data are added to the manuscript in Figures 2 and 3.

*2) The current manuscript is mainly focusing on excitatory synapses situated in dendritic spines, but lacks data for inhibitory synapses. The authors need to visualize inhibitory synaptic markers such as VGAT and gephyrin (*Figure 1*), and measure mIPSCs for some of the results described in*
Figures 3, 4 and 5.

Gamma-secretase complexes cleave their target subsequently to alfa- or beta-secretase cleavages of the ectodomain. As the ectodomain is in principle already shed we do not anticipate that forward signalling of their target proteins will be affected by downstream gamma-secretase cleavage. Consistent with this model, our previous study suggested that gamma-secretase processing does not impact on Nrg1 forward signalling (13). This observation predicts that inhibitory synapses should not be altered in gamma-secretase mutant mice since inhibitory synapses are regulated by ErbB4 activation.

In order to test this experimentally as suggested by the reviewers, we visualized inhibitory boutons with the markers VGAT, which labels pre-synaptically all inhibitory synapses, and with PV, since PV+ interneurons are the subpopulation in interneurons where ErbB4 is primarily expressed ([17]; Wen L et al., 2010). The intensities of VGAT and PV immunoreactive puncta did not reveal any difference between control and Aph1BC mutant mice. These data are added as Figure 1—figure supplement 1. Considering this observation and the facts that Nrg1 is exclusively expressed by pyramidal neurons and that our study is focused on the role of Aph1BC-gamma-secretase/Nrg1 intracellular signalling in excitatory synapses, we consider that the measurement of mIPSCs would be redundant and out of the scope of this manuscript.

*Minor comments*:

*1) The Sholl analysis shown in*
Figure 5
*seems to analyze only the proximal regions of the dendrites (∼50 microns). But the dendrites of the CA1 neurons, especially the apical dendrites, are much longer, extending several hundred microns. The coverage should be increased. In addition, the SLM layer of CA1 is not indicated in the figure*.

Pyramidal neuron in the hippocampus display two kind of dendritic arborisations: the basal dendrites extend themselves in the stratum oriens and display a short and thick arborisation; conversely, the apical dendrites extend themselves in the stratum radiatum with a much longer and less dense arborisation pattern. According to reviewer’s indication, quantification was extended to a longer range and for sake of clarity and completeness analysis of basal and apical dendrites are presented separately. We add this novel figure in Figure 6. SLM is indicated in Figure 6.

*2) Given the majority of the experiments were performed in the hippocampus, the title of the manuscript containing “…cortical excitatory circuitry…” should be toned down*.

We changed the title to “hippocampal excitatory circuitry”.

*3) In the Abstract, the authors state that “…Aph1B, but not Aph1A, is selectively involved in NRG1 intracellular signaling”, but the results in*
Figure 3
*implicate Aph1A in the processing of NRG1. This should be discussed*.

The Abstract was modified and “but not Aph1A” was deleted. A comment to clarify our view on the selective role of Aph1A- vs Aph1BC-gamma-secretase complexes was also added in the discussion. We wrote in the Discussion: “Besides, even though Aph1A-γ-secretase might redundantly contribute to Nrg1 processing, collectively our data indicate that Aph1BC-γ-secretase is the major regulator of Nrg1 intracellular signalling in this biological process.”

*4) All three panels in*
Figure 2
*lack sample current traces*.

Sample current traces were added as requested in Figure 2.

*5) The reduced LTP in Aph1BC-/- neurons (*Figure 2*) could be caused by the reduced basal transmission (*Figure 2*). This possibility should be mentioned*.

In the protocol that we used, the I/O curve is measured first. LTP is subsequently induced by a stimulation of 35% of the maximum (i.e. saturating) stimulation. This procedure thus allows accounting for the variability between slices and genotypes in the measurement of long term plasticity. Therefore, we consider it unlikely that LTP deficiency might be secondary to a deficit in baseline transmission. Nevertheless the referee is right that this experimental paradigm may be in principle biased by reduced basal transmission and this possibility is now mentioned: “Even though reduced basal transmission might in principle interfere with LTP analysis, these results suggest nevertheless that synaptic plasticity is affected by Aph1BC deletion.”

*6) In the Discussion, the authors appear to challenge previous work showing that impairment of neuregulin 1 intracellular formation leads to impaired dendritic arborisation (Chen et al., 2010), given that they failed to observe differences in dendritic length in the current study. However, the difference in results may be due to long-term effects of impaired NRG1 intracellular signaling, which is not suffered in the conditional deletion studies. This possibility should be acknowledged*.

We thank the reviewers for this comment that prompts us to better express and clarify this point. The effect of Nrg1 loss of function on dendritic arborisation was previously studied in two interesting papers from the group of Talmage and Role (quoted references 8 and 32). We do not believe that our study challenges the work by Chen et al, 2010 but indeed is consistent with the observation of Talmage and Role that Nrg1 heterozygous mice do not display abnormalities in dendritic arborisation in vivo. The diverse timeline of abrogation of Nrg1 intracellular signalling in constitutive vs conditional studies is a reasonable explanation for this apparent discrepancy and we acknowledge this now in the Discussion: “We speculate that these differences may be explained by differential effects of long term abrogation of Nrg1 intracellular signalling in Nrg1 constitutive null mice which is not the case in *Aph1BC*^*-/-*^ conditional mice.”